# INITIALIZATION MATTERS: ORTHOGONAL PREDICTIVE STATE RECURRENT NEURAL NETWORKS

**Krzysztof Choromanski**[*]
Google Brain
kchoro@google.com

**Carlton Downey**[*†]
Carnegie Mellon University
cmdowney@cs.cmu.edu

**Byron Boots**[‡]
Georgia Tech
bboots@cc.gatech.edu

## ABSTRACT

Learning to predict complex time-series data is a fundamental challenge in a range of disciplines including Machine Learning, Robotics, and Natural Language Processing. Predictive State Recurrent Neural Networks (PSRNNs) (Downey et al., 2017) are a state-of-the-art approach for modeling time-series data which combine the benefits of probabilistic filters and Recurrent Neural Networks into a single model. PSRNNs leverage the concept of Hilbert Space Embeddings of distributions (Smola et al., 2007) to embed predictive states into a Reproducing Kernel Hilbert Space, then estimate, predict, and update these embedded states using Kernel Bayes Rule. Practical implementations of PSRNNs are made possible by the machinery of Random Features, where input features are mapped into a new space where dot products approximate the kernel well. Unfortunately PSRNNs often require a large number of RFs to obtain good results, resulting in large models which are slow to execute and slow to train. Orthogonal Random Features (ORFs)(Yu et al., 2016) is an improvement on RFs which has been shown to decrease the number of RFs required for pointwise kernel approximation. Unfortunately, it is not clear that ORFs can be applied to PSRNNs, as PSRNNs rely on Kernel Ridge Regression as a core component of their learning algorithm, and the theoretical guarantees of ORF do not apply in this setting. In this paper, we extend the theory of ORFs to Kernel Ridge Regression and show that ORFs can be used to obtain Orthogonal PSRNNs (OPSRNNs), which are smaller and faster than PSRNNs. In particular, we show that OPSRNN models clearly outperform LSTMs and furthermore, can achieve accuracy similar to PSRNNs with an order of magnitude smaller number of features needed.

## 1 INTRODUCTION

Learning to predict temporal sequences of observations is a fundamental challenge in a range of disciplines including machine learning, robotics, and natural language processing. There exist a wide variety of approaches to modeling time series data, however recurrent neural networks (RNNs) have emerged as the clear frontrunner, achieving state-of-the-art performance in applications such as speech recognition (Heigold et al., 2016), language modeling (Mikolov et al., 2010), translation (Cho et al., 2014b), and image captioning (Xu et al., 2015).

Predictive State Recurrent Neural Networks (PSRNNs) are a state-of-the-art RNN architecture recently introduced by Downey et al. (2017) that combine the strengths of probabilistic models and RNNs in a single model. Specifically PSRNNs offer strong statistical theory, globally consistent model initializations, and a rich functional form which is none-the-less amenable to refinement via backpropagation through time (BPTT). Despite consisting of a simple bi-linear operations, PSRNNs have been shown to significantly outperform more complex RNN architectures (Downey et al., 2017), such as the widely used LSTMs (Hochreiter & Schmidhuber, 1997) and GRUs (Cho et al., 2014a).

---

[*]Equal Contribution
[†]Work done while at Google
[‡]Work done while at Google Brain

PSRNNs leverage the concept of Hilbert Space embeddings of distributions (Smola et al., 2007) to embed predictive states into a Reproducing Kernel Hilbert Space (RKHS), then estimate, predict, and update these embedded states using Kernel Bayes Rule (KBR) (Smola et al., 2007). Because PSRNNs directly manipulate (kernel embeddings of) distributions over observations, they can be initialized via a globally consistent method-of-moments algorithm which reduces to a series of linear ridge regressions.

Practical implementations of PSRNNs are made possible by the machinery of Random Features (RFs): input features are mapped into a new space where dot products approximate the kernel well (Rahimi & Recht, 2008). RFs are crucial to the success of PSRNNs, however PSRNNs often require a significant number of RFs in order to obtain good results. And, unfortunately, the number of required RFs grows with the dimensionality of the input, resulting in models which can be large, slow to execute, and slow to train.

One technique that has proven to be effective for reducing the required number of RFs for kernel machines is Orthogonal Random Features (ORFs) (Yu et al., 2016). When using ORFs, the matrix of RFs is replaced by a properly scaled random orthogonal matrix, resulting in significantly decreased kernel approximation error. A particularly nice feature of ORFs is that (Yu et al., 2016; Choromanski et al., 2017) prove that using ORFs results in a guaranteed improvement in pointwise kernel approximation error when compared with RFs.

Unfortunately the guarantees in Yu et al. (2016) are not directly applicable to the PSRNN setting. PSRNNs first obtain a set of model parameters via ridge regression, then use these model parameters to calculate inner products in RF space. This "downstream" application of RFs goes beyond the results proven in Yu et al. (2016) and Choromanski et al. (2017). Hence it is not clear whether or not ORF can be applied to obtain an improvement in the PSRNN setting.

In this work, we show that ORFs can be used to obtain OPSRNNs: PSRNNs initialized using ORFs which are smaller, faster to execute and train than PSRNNs initialized using conventional unstructured RFs. We theoretically analyze the orthogonal version of the KRR algorithm that is used to initialize OPSRNNs. We show that orthogonal RNNs lead to kernel algorithms with strictly better spectral properties and explain how this translates to strictly smaller upper bounds on failure probabilities regarding KRR empirical risk. We compare the performance of OPSRNNs with that of LSTMs as well as conventional PSRNNs on a number of robotics tasks, and show that OPSRRNs are consistently superior on all tasks. In particular, we show that OPSRNN models can achieve accuracy similar to PSRNNs with an order of magnitude smaller number of features needed.

## 2  RELATED WORK

Orthogonal random features were introduced in Yu et al. (2016) as an alternative to the standard approach for constructing random feature maps to scale kernel methods (Rahimi & Recht, 2007). Several other structured constructions were known before (Ailon & Chazelle, 2006; Hinrichs & Vybíral, 2011; Vybíral, 2011; Zhang & Cheng, 2013; Choromanski & Sindhwani, 2016; Choromanska et al., 2016; Bojarski et al., 2017), however these were motivated by computational and space complexity gains and led to weaker accuracy guarantees. In contrast to this previous work, orthogonal random features were proposed to improve accuracy of the estimators relying on them. Such an improvement was theoretically and experimentally verified, but only for pointwise kernel approximation (Yu et al., 2016; Choromanski et al., 2017) and for specific kernels (such as Gaussian for dimensionality large enough, as well as dot-product and angular kernels). It was not clear whether these pointwise gains translate to downstream guarantees for algorithms relying on kernels (for instance kernel ridge regression), even though there was some partial empirical evidence that this might be the case (in Choromanski et al. (2017) orthogonal random features were experimentally tested to provide more accurate approximation of the groundtruth kernel matrix in terms of the Frobenius norm error). Even for the pointwise estimators and for the selected kernels, the guarantees were given only with the use of second moment methods (variance computation) and thus did not lead to strong concentration results with exponentially small probabilities of failure, which we obtain in this paper.

To the best of our knowledge, we are the first to apply orthogonal random features via kernel ridge regression for recurrent neural networks. There is however vast related literature on orthogonal

recurrent neural networks, where the matrices are enforced to be orthogonal or initialized to be random orthogonal. Probably some of the most exploited recent directions are unitary evolution RNN architectures (Arjovsky et al., 2016), where orthogonal and unitary matrices are used to address a key problem of RNN training - vanishing or exploding gradients. Related results are presented in Henaff et al. (2016), Saxe et al. (2013) (orthogonal initialization for training acceleration), Ganguli et al. (2008) and White et al. (2004). Most of these results do not provide any strict theoretical guarantees regarding the superiority of the orthogonal approach. Even though these approaches are only loosely related to our work, there is a common denominator: orthogonality, whether applied in our context or the aforementioned ones, seems to to be responsible for disentangling in (deep) representations (Achille & Soatto, 2017). Our rigorous theoretical analysis shows that this phenomenon occurs for the orthogonal KRR that is used as a subroutine of OPSRNNs, but the general mechanism is still not completely understood from the theoretical point of view.

## 3 PREDICTIVE STATE RECURRENT NEURAL NETWORKS

PSRNNs (Downey et al., 2017) are a recently developed RNN architecture which combine the ideas of predictive state (Boots et al., 2013) and RFs. The key advantage of PSRNNs is that their state update can be interpreted in terms of Bayes Rule. This allows them to be initialized as a fully functional model via a consistent method of moments algorithm, in contrast to conventional RNN architectures which are initialized at random. Below we describe PSRNNs in more detail. We pay particular attention to how PSRNNS utilize RFs, which will be replaced with ORFs in OPSRNNs. The explicit construction of ORFs is given in Section 4.

A predictive state (Littman et al., 2001) is a set of expectations of features of future observations. A predictive state is best understood in contrast with the more typical latent state: predictive states are distributions over known functions of observations, whereas latent states are distributions over unobserved latent quantities. Formally, a predictive state is a vector $q_t = E[f_t \mid h_t]$, where $f_t = f(o_{t:t+k-1})$ are features of future observations and $h_t = h(o_{1:t-1})$ are features of historical observations.

PSRNNs use a predictive state, but embed it in a Reproducing Kernel Hilbert Space (RKHS) using RFs: Let $k_f$, $k_h$, $k_o$ be translation invariant kernels (Rahimi & Recht, 2008) defined on $f_t$, $h_t$, and $o_t$ respectively. Define projections $\phi_t = RF(f_t)$, $\eta_t = RF(h_t)$, and $\omega_t = RF(o_t)$ such that $k_f(f_i, f_j) = \phi_i^T \phi_j$, $k_h(h_i, h_j) = \eta_i^T \eta_j$, $k_o(o_i, o_j) = \omega_i^T \omega_j$. Then the PSRNN predictive state is $q_t = E[\phi_t \mid \eta_t]$.

PSRNN model parameters consist of an initial state $q_1$ and a 3-mode update tensor $W$. The PSRNN state update equation is:

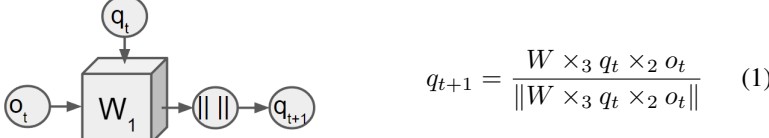

$$q_{t+1} = \frac{W \times_3 q_t \times_2 o_t}{\|W \times_3 q_t \times_2 o_t\|} \quad (1)$$

Figure 1: PSRNN Update, displayed on the left as a neural network and on the right as an equation

### 3.1 TWO-STAGE REGRESSION FOR PSRNNS

PSRNNs can be initialized using the Two Stage Regression (2SR) approach of Hefny et al. (2015). This approach is fast, statistically consistent, and reduces to simple linear algebra operations. In 2SR $q_1$ and $W$ are learned by solving three Kernel Ridge Regression problems in two stages. Ridge regression is required in order to obtain a stable model, as it allows us to minimize the destabilizing effect of rare events while preserving statistical consistency.

In stage one we regress from past $\phi_t$ to future $\eta_t$, and from past $\phi_t$ to the outer product of shifted future $\psi := \eta_{t+1}$ with observations $\omega_t$. Let $X_\phi$ be the matrix whose $t$th column is $\phi_t$, $X_\eta$ the matrix whose $t$th column is $\eta_t$, and $X_{\psi \otimes \omega}$ be the matrix whose $t$th column is $\psi_t \otimes \omega_t$:

$$W_{\eta|\phi} = X_\eta X_\phi^T \left( X_\phi X_\phi^T + \lambda I \right)^{-1},$$
$$W_{\psi\omega|\phi} = X_{\psi\otimes\omega} X_\phi^T \left( X_\phi X_\phi^T + \lambda I \right)^{-1}.$$

Using $W_{\eta|\phi}$ and $W_{\psi\omega|\phi}$ we obtain predictive state estimates $X_{\eta|\phi}$ and $X_{\psi\omega|\phi}$ at each time step:

$$X_{\eta|\phi} = W_{\eta|\phi} X_\phi$$
$$X_{\psi\omega|\phi} = W_{\psi\omega|\phi} X_\phi$$

In stage two we regress from $C_{\eta|\phi}$ to $C_{\psi\omega|\phi}$ to obtain the model weights $W$:

$$q_1 = X_{\eta|\phi}\mathbf{1},$$
$$W = X_{\psi\omega|\phi} X_{\eta|\phi}^T \left( X_{\eta|\phi} X_{\eta|\phi}^T + \lambda I \right)^{-1},$$

where $\lambda \in \mathbb{R}$ is the ridge parameter and $I$ is the identity matrix and $\mathbf{1}$ is a column vector of ones. Once the state update parameters have been learned via 2SR we train a kernel ridge regression model to predict $\omega_t$ from $q_t$. [1]

The 2SR algorithm is provably consistent in the realizable setting, meaning that in the limit we are guaranteed to recover the true model parameters. Unfortunately this result relies on exact kernel values, while scalable implementations work with approximate kernel values via the machinery of RFs. In practice we often require a large number of RFs in order to obtain a useful model. This can result in large models which are slow to execute and slow to train.

We now introduce ORFs and show that they can be used to obtain smaller, faster models. The key challenge with applying ORFs to PSRNNs is extending the theoretical guarantees of Yu et al. (2016) to the kernel ridge regression setting.

## 4 ORTHOGONAL RANDOM FEATURES

We explain here how to construct orthogonal random features to approximate values of kernels defined by the prominent family of radial basis functions and consequently, conduct kernel ridge regression for the OPSRNN model. A class of RBF kernels $\mathcal{K}$ (RBFs in shorthand) is a family of functions: $K_n : \mathbb{R}^n \times \mathbb{R}^n \to \mathbb{R}$ for $n = 1, 2, ...$ such that $K_n(\mathbf{x}, \mathbf{y}) = \phi(z)$, for $z = \|\mathbf{x} - \mathbf{y}\|_2$, where $\phi : \mathbb{R} \to \mathbb{R}$ is a fixed positive definite function (not parametrized by $n$). An important example is the class of Gaussian kernels.

Every RBF kernel $K$ is shift-invariant, thus in particular its values can be described by an integral via Bochner's Theorem (Rahimi & Recht, 2007):

$$K(\mathbf{x}, \mathbf{y}) = \mathrm{Re} \int_{\mathbb{R}^n} \exp(i\mathbf{w}^\top(\mathbf{x} - \mathbf{y}))\mu_K(d\mathbf{w}), \tag{2}$$

where $\mu_K \in \mathcal{M}(\mathbb{R}^n)$ stands for some finite Borel measure. Some commonly used RBF kernels $K$ together with the corresponding functions $\phi$ and probability density functions for measures $\mu_K$ are given in Table 2. The above formula leads straightforwardly to the standard unbiased Monte-Carlo estimator of $K(\mathbf{x}, \mathbf{y})$ given as: $K(\mathbf{x}, \mathbf{y}) = \Phi_{m,n}^\top(\mathbf{x})\Phi_{m,n}(\mathbf{y})$, where a random embedding $\Phi_{m,n} : \mathbb{R}^n \to \mathbb{R}^{2m}$ is given as:

$$\Phi_{m,n}(\mathbf{x}) = \left( \frac{1}{\sqrt{m}} \cos(\mathbf{w}_i^\top \mathbf{x}), \frac{1}{\sqrt{m}} \sin(\mathbf{w}_i^\top \mathbf{x}) \right)_{i=1}^m, \tag{3}$$

vectors $\mathbf{w}_i \in \mathbb{R}^n$ are sampled independently from $\mu_K$ and $m$ stands for the number of random features used. In this scenario we will often use the notation $\mathbf{w}_i^{\mathrm{iid}}$, to emphasize that different $\mathbf{w}_i$s are sampled independently from the same distribution.

---

[1] Note that we can train a regression model to predict any quantity from the state

| Name | Positive-definite function $\phi$ | Probability density function |
|------|-----------------------------------|------------------------------|
| Gaussian | $\sigma^2 \exp\left(-\dfrac{1}{2\lambda^2} z^2\right)$ | $\frac{\sigma^2}{(2\pi\lambda^2)^{n/2}} \exp\left(-\frac{1}{2\lambda^2}\|\mathbf{w}\|_2^2\right)$ |
| Laplacian | $\exp(-z)$ | $\prod_{i=1}^{n} \frac{1}{\pi(1+w_i)}$ |

Figure 2: Common RBF kernels, the corresponding functions $\phi$, and probability density functions (here: $\mathbf{w} = (w_1, ..., w_n)^\top$).

For a datasets $\mathcal{X}$, random features provide an equivalent description of the original kernel via the linear kernel in the new dataset $\Phi(\mathcal{X}) = \{\Phi(\mathbf{x}) : \mathbf{x} \in \mathcal{X}\}$ obtained by the nonlinear transformation $\Phi$ and lead to scalable kernel algorithms if the number of random features needed to accurately approximate kernel values satisfies: $m \ll N = |\mathcal{X}|$.

Orthogonal random features are obtained by replacing a standard mechanism of constructing vectors $\mathbf{w}_i$ and described above with the one where the sequence $(\mathbf{w}_1, ..., \mathbf{w}_m)$ is sampled from a "related" joint distribution $\mu_{K,m,n}^{\mathrm{ort}}$ on $R^n \times ... \times R^n$ satisfying the orthogonality condition, namely: with probability $p = 1$ different vectors $\mathbf{w}_i$ are pairwise orthogonal. Since in practice we often need $m > n$, the sequence $(\mathbf{w}_i)_{i=1,...,m}$ is obtained by stacking some number of blocks, each of length $l \leq n$, where the blocks are sampled independently from $\mu_{K,m,n}^{\mathrm{ort}}$.

It remains to explain how $\mu_{K,m,n}^{\mathrm{ort}}$ is constructed. We consider two main architectures. For the first one the marginal distributions (distributions of individual vectors $\mathbf{w}_i$) are $\mu_K$. A sample $(\mathbf{w}_1^{\mathrm{ort}}, ..., \mathbf{w}_m^{\mathrm{ort}})$ from the joint distribution might be obtained for instance by constructing a random matrix $\mathbf{G} = [(\mathbf{w}_1^{\mathrm{iid}})^\top; ...; (\mathbf{w}_m^{\mathrm{iid}})^\top] \in \mathbb{R}^{m \times n}$, performing Gram-Schmidt orthogonalization and then renormalizing the rows such that the length of the renormalized row is sampled from the distribution from which $\|\mathbf{w}_i^{\mathrm{iid}}\|$s are sampled. Thus the Gram-Schmidt orthogonalization is used just to define the directions of the vectors. From now on, we will call such a joint distribution *continuous-orthogonal*. The fact that for RBF kernels the marginal distributions are exactly $\mu_K$ and thus, kernel estimator is still unbiased, is a direct consequence of the isotropicity of distributions fom which directions of vectors $\mathbf{w}_i^{\mathrm{iid}}$ are sampled. For this class of orthogonal estimators we prove strong theoretical guarantees showing that they lead to kernel ridge regression models superior to state-of-the-art ones based on vectors $\mathbf{w}_i^{\mathrm{iid}}$.

Another class of orthogonal architectures considered by us is based on discrete matrices. We denote by $\mathbf{D}$ a random diagonal matrix with nonzero entries taken independently and uniformly at random from the two-element set $\{-1, +1\}$. Furthermore, we will denote by $\mathbf{H}$ a *Hadamard matrix* obtained via Kronecker-products (see: Choromanski et al. (2017)). An $m$-vector sample from the *discrete-orthogonal* joint distribution is obtained by taking $m$ first rows of matrix $\mathbf{G}$ defined as $\mathbf{G}_{\mathrm{HAD}} = \mathbf{H}\mathbf{D}_1 \cdot ... \cdot \mathbf{H}\mathbf{D}_k$ for: fixed $k > 0$, independent copies $\mathbf{D}_i$ of $\mathbf{D}$ and then renormalizing each row in exactly the same way as we did it for continuous-orthogonal joint distributions. Note that $\mathbf{G}_{\mathrm{HAD}}$ is a product of orthogonal matrices, thus its rows are also orthogonal. The advantage of a discrete approach is that it leads to a more time and space efficient method for computing random feature maps (with the use of Fast Walsh-Hadamard Transform; notice that the Hadamard matrix does not even need to be stored explicitly). This is not our focus in this paper though. Accuracy-wise discrete-orthogonal distributions lead to slightly biased estimators (the bias is a decreasing function of the dimensionality $n$). However as we have observed, in practice they give as accurate PSRNN models as continuous-orthogonal distributions, consistently beating approaches based on unstructured random features. One intuitive explanation of that phenomenon is that even though in that setting kernel estimators are biased, they are still characterized by much lower variance than those based on unstructured features. We leave a throughout theoretical analysis of discrete-orthogonal joint distributions in the RNN context to future work.

## 5    THE THEORY OF THE ORTHOGONAL KERNEL RIDGE REGRESSION

In this section we extend the theoretical guarantees of Yu et al. (2016) to give rigorous theoretical analysis of the initialization phase of OPSRNN. Specifically, we provide theoretical guarantees for kernel ridge regression with orthogonal random features, showing that they provide strictly better spectral approximation of the ground-truth kernel matrix than unstructured random features. As a

corollary, we prove that orthogonal random features lead to strictly smaller empirical risk of the model. Our results go beyond second moment guarantees and enable us to provide the first exponentially small bounds on the probability of a failure for random orthogonal transforms.

Before we state our main results, we will introduce some basic notation and summarize previous results. Assume that labeled datapoints $(\mathbf{x}_i, y_i)$, where $\mathbf{x}_i \in \mathbb{R}^n$, $y_i \in \mathbb{R}$ for $i = 1, 2, ...$, are generated as follows: $y_i = f^*(\mathbf{x}_i) + \nu_i$, where $f^* : \mathbb{R}^n \to \mathbb{R}$ is a function that the model aims to learn, and $\nu_i$ for $i = 1, 2, ...$ are independent Gaussians with zero mean and standard deviation $\sigma > 0$. The *empirical risk of the estimator* $f : \mathbb{R}^n \to \mathbb{R}$ is defined as follows:

$$\mathcal{R}(f) \equiv \mathbb{E}_{\{\nu_i\}_{i=1,...,N}}\big[\frac{1}{N}\sum_{j=1}^{N}(f(\mathbf{x}_i) - f^*(\mathbf{x}_i))^2\big], \tag{4}$$

where $N$ stands for a dataset size.

By $\mathbf{f}_{\mathrm{vec}}^* \in \mathbb{R}^N$ we denote a vector whose $j^{th}$ entry is $f^*(\mathbf{x}_j)$. Denote by $f_{\mathrm{KRR}}$ a kernel ridge regression estimator applying exact kernel method (no random feature map approximation). Assume that we analyze kernel $K : \mathbb{R}^n \times \mathbb{R}^n \to \mathbb{R}$ with the corresponding kernel matrix $\mathbf{K}$. It is a well known result (Alaoui & Mahoney, 2015; Bach, 2013) that the empirical risk of $f_{\mathrm{KRR}}$ is given by the formula:

$$\mathcal{R}(f_{\mathrm{KRR}}) = N^{-1}\lambda^2(\mathbf{f}_{\mathrm{vec}}^*)^\top(\mathbf{K} + \lambda N\mathbf{I}_N)^{-2}\mathbf{f}_{\mathrm{vec}}^* + N^{-1}\sigma^2\mathrm{Tr}(\mathbf{K}^2(\mathbf{K} + \lambda N\mathbf{I}_N)^{-2}), \tag{5}$$

where $\lambda > 0$ stands for the regularization parameter and $\mathbf{I}_N \in \mathbb{R}^{N \times N}$ is an identity matrix.

Denote by $\widehat{f}_{\mathrm{KRR}}$ an estimator based on some random feature map mechanism and by $\widehat{\mathbf{K}}$ the corresponding approximate kernel matrix.

The expression that is used in several bounds on the empirical risk for kernel ridge regression (see for instance Avron et al. (2017)) is the modified version of the above formula for $\mathcal{R}(f_{\mathrm{KRR}})$, namely: $\bar{\mathcal{R}}_{\mathbf{K}}(\mathbf{f}_{\mathrm{vec}}^*) \equiv N^{-1}\lambda^2(\mathbf{f}_{\mathrm{vec}}^*)^\top(\mathbf{K} + \lambda N\mathbf{I}_N)^{-1}\mathbf{f}_{\mathrm{vec}}^* + N^{-1}\sigma^2 s_\lambda(\mathbf{K})$, where $s_\lambda(\mathbf{K}) \equiv \mathrm{Tr}(\mathbf{K}(\mathbf{K} + \lambda N\mathbf{I}_N)^{-1})$. It can be easily proven that $\mathcal{R}(f_{\mathrm{KRR}}) \leq \bar{\mathcal{R}}_{\mathbf{K}}(\mathbf{f}_{\mathrm{vec}}^*)$.

To measure how similar to the exact kernel matrix (in terms of spectral properties) a kernel matrix obtained with random feature maps is, we use the notion of $\Delta$-spectral approximation (Avron et al., 2017).

**Definition 1.** *For a given $0 < \Delta < 1$, matrix $\mathbf{A} \in \mathbb{R}^{N \times N}$ is a $\Delta$-spectral approximation of a matrix $\mathbf{B} \in \mathbb{R}^{N \times N}$ if $(1 - \Delta)\mathbf{B} \preceq \mathbf{A} \preceq (1 + \Delta)\mathbf{B}$.*

It turns out that one can upper-bound the risk $\mathcal{R}(\widehat{f}_{\mathrm{KRR}})$ for the estimator $\widehat{f}_{\mathrm{KRR}}$ in terms of the $\Delta$ parameter if matrix $\widehat{\mathbf{K}} + \lambda N\mathbf{I}_N$ is a $\Delta$-spectral approximation of the matrix $\mathbf{K} + \lambda N\mathbf{I}_N$, as the next result (Avron et al., 2017) shows:

**Theorem 1.** *Suppose that $\|\mathbf{K}\|_2 \geq 1$ and that matrix $\widehat{\mathbf{K}} + \lambda N\mathbf{I}_N$ obtained with the use of random features is a $\Delta$-spectral approximation of matrix $\mathbf{K} + \lambda N\mathbf{I}_N$. Then the empirical risk $\mathcal{R}(\widehat{f}_{\mathrm{KRR}})$ of the estimator $\widehat{f}_{\mathrm{KRR}}$ satisfies:*

$$\mathcal{R}(\widehat{f}_{\mathrm{KRR}}) \leq \frac{1}{1 - \Delta}\bar{\mathcal{R}}_{\mathbf{K}}(\mathbf{f}_{\mathrm{vec}}^*) + \frac{\Delta}{1 + \Delta}\frac{\mathrm{rank}(\widehat{\mathbf{K}})}{N}\sigma^2. \tag{6}$$

## 5.1 SUPERIORITY OF THE ORTHOGONAL FEATURES FOR KERNEL RIDGE REGRESSION

Consider the following RBF kernels, that we call *smooth RBFs*. As we show next, Gaussian kernels are smooth.

**Definition 2** (smooth RBFs). *We say that the class of RBF kernels defined by a fixed $\phi : \mathbb{R} \to \mathbb{R}$ (different elements of the class corresponds to different input dimensionalities) and with associated sequence of probabilistic measures $\{\mu_1, \mu_2, ...\}$ ($\mu_i \in \mathcal{M}(\mathbb{R}^i)$) is smooth if there exists a*

*nonincreasing function* $f : \mathbb{R} \rightarrow \mathbb{R}$ *such that* $f(x) \rightarrow 0$ *as* $x \rightarrow \infty$ *and furthermore the* $k^{th}$ *moments of random variables* $X_n = \|\mathbf{w}\|$, *where* $\mathbf{w} \sim \mu_n$ *satisfy for every* $n, k \geq 0$: $\mathbb{E}[X_n^k] \leq (n-1)(n+1) \cdot \ldots \cdot (n+2k-3)k! f^k(k)$.

Many important classes of RBF kernels are smooth, in particular the class of Gaussian kernels. This follows immediately from the well-known fact that for Gaussian kernels the above $k^{th}$ moments are given by the following formula: $\mathbb{E}[X_n^k] = 2^k \frac{(\frac{n}{2}+k-1)!}{(\frac{n}{2}-1)!}$ for $n > 1$.

Our main result is given below and shows that orthogonal random features lead to tighter bounds on $\Delta$ for the spectral approximation of $\mathbf{K} + \lambda N \mathbf{I}_N$. Tighter bounds on $\Delta$, as Theorem 1 explains, lead to tighter upper bounds also on the empirical risk of the estimator. We will prove it for the setting where each structured block consists of a fixed number $l > 1$ of rows (note that many independent structured blocks are needed if $m > n$), however our experiments suggest that the results are valid also without this assumption.

**Theorem 2** (spectral approximation). *Consider a smooth RBF (in particular Gaussian kernel). Let* $\widehat{\Delta}_{\mathrm{iid}}$ *denote the smallest positive number such that* $\widehat{\mathbf{K}}_{\mathrm{iid}} + \lambda N \mathbf{I}_N$ *is a* $\Delta$-*approximation of* $\mathbf{K} + \lambda N \mathbf{I}_N$, *where* $\widehat{\mathbf{K}}_{\mathrm{iid}}$ *is an approximate kernel matrix obtained by using unstructured random features. Then for any* $a > 0$,

$$\mathbb{P}[\widehat{\Delta}_{\mathrm{iid}} > a] \leq p_{N,m}^{\mathrm{iid}}\left(\frac{a\sigma_{\min}}{N}\right), \tag{7}$$

*where:* $p_{N,m}^{\mathrm{iid}}$ *is given as:* $p_{N,m}^{\mathrm{iid}}(x) = N^2 e^{-Cmx^2}$ *for some universal constant* $C > 0$, *m is the number of random features used,* $\sigma_{\min}$ *is the smallest singular value of* $\mathbf{K} + \lambda N \mathbf{I}_N$ *and N is dataset size. If instead orthogonal random features are used then for the corresponding spectral parameter* $\widehat{\Delta}_{\mathrm{ort}}$ *the following holds:*

$$\mathbb{P}[\widehat{\Delta}_{\mathrm{ort}} > a] \leq p_{N,m}^{\mathrm{ort}}\left(\frac{a\sigma_{\min}}{N}\right), \tag{8}$$

*where function* $p_{N,m}^{\mathrm{ort}}$ *satisfies:* $p_{N,m}^{\mathrm{ort}} < p_{N,m}^{\mathrm{iid}}$ *for n large enough.*

We see that both constructions lead to exponentially small (in the number of random features $m$ used) probabilities of failure, however the bounds are tighter for the orthogonal case. An exact formula on $p_{N,m}^{\mathrm{ort}}$ can be derived from the proof that we present in the Appendix, however for clarity we do not give it here.

Theorem 2 combined with Theorem 1 lead to risk bounds for the kernel ridge regression model based on random unstructured and random orthogonal features. We use the notation introduced before and obtain the following:

**Theorem 3.** *Under the assumptions of Theorem 1 and Theorem 2, the following holds for the kernel ridge regression risk and any* $c > 0$ *if* $m$-*dimensional unstructured random feature maps are used to approximate a kernel:* $\mathbb{P}[\mathcal{R}(\widehat{f}_{\mathrm{KRR}}) > c] \leq p_{N,m}^{\mathrm{iid}}\left(\frac{a_c\sigma_{\min}}{N}\right)$, *where* $a_c$ *is given as:* $a_c = 1 - \frac{\overline{\mathcal{R}}_{\mathbf{K}}(\mathbf{f}_{\mathrm{vec}}^*)}{c - \frac{m\sigma^2}{2N}}$ *and the probability is taken with respect to the random choices of features. If instead random orthogonal features are used, we obtain the following bound:* $\mathbb{P}[\mathcal{R}(\widehat{f}_{\mathrm{KRR}}) > c] \leq p_{N,m}^{\mathrm{ort}}\left(\frac{a_c\sigma_{\min}}{N}\right)$.

As before, since for large $n$ function $p_{N,m}^{\mathrm{ort}}$ satisfies $p_{N,m}^{\mathrm{ort}} < p_{N,m}^{\mathrm{iid}}$, for orthogonal random features we obtain strictly smaller upper bounds on the failure probability regarding empirical risk than for the state-of-the-art unstructured ones. In practice, as we will show in the experimental section, we see gains also in the regimes of moderate dimensionalities $n$.

## 6 EXPERIMENTS

In section 5 we extended the theoretical guarantees for ORFs to the case of the initialization phase of OPSRNNs. In this section we confirm these results experimentally and show that they imply better performance of the entire model by comparing the performance of PSRNNs with that of OPSRNNs on a selection of robotics time-series datasets. Since OPSRNN models obtained via continuous-orthogonal and discrete-orthogonal joint sampling (see: Section 4) gave almost the same results, presented OPSRNN-curves are for the continuous-orthogonal setting.

## 6.1 EXPERIMENTAL SETUP

We now describe the datasets and model hyperparameters used in our experiments. All models were implemented using the Tensorflow framework in Python.

We use the following datasets in our experiments:

- **Swimmer** We consider the 3-link simulated swimmer robot from the open-source package OpenAI gym.[2]. The observation model returns the angular position of the nose as well as the (2D) angles of the two joints, giving in a total of 5 features. We collect 25 trajectories from a robot that is trained to swim forward (via the cross entropy with a linear policy), with a train/test split of 20/5.
- **Mocap** A Human Motion Capture dataset consisting of 48 skeletal tracks from three human subjects collected while they were walking. The tracks have 300 time steps each, and are from a Vicon motion capture system. We use a train/test split of 40/8. There are 22 total features consisting of the 3D positions of the skeletal parts (e.g., upper back, thorax, clavicle).
- **Handwriting** This is a digital database available on the UCI repository (Alpaydin & Alimoglu, 1998) created using a pressure sensitive tablet and a cordless stylus. Features are $x$ and $y$ tablet coordinates and pressure levels of the pen at a sampling rate of 100 milliseconds giving a total of 3 features. We use 25 trajectories with a train/test split of 20/5.
- **Moving MNIST** Pairs of MNIST digits bouncing around inside of a box according to ideal physics. `http://www.cs.toronto.edu/~nitish/unsupervised_video/`. Each video is 64x64 pixels single channel (4096 features) and 20 frames long. We use 1000 randomly selected videos, split evenly between train and test.

In two-stage regression we use history (similarly future) features consisting of the past (next) 2 observations concatenated together. We use a ridge-regression parameter of $10^{(-2)}$ (this is consistent with the values suggested in Boots et al. (2013); Downey et al. (2017)). The kernel width is set to the median pairwise (Euclidean) distance between neighboring data points. We use a fixed learning rate of $0.1$ for BPTT with a BPTT horizon of 20. We use a single layer PSRNN.

We optimize and evaluate all models with respect to the Mean Squared Error (MSE) of one step predictions (this should not be confused with the MSE of the pointwise kernel approximation which does not give the downstream guarantees we are interested in here). This means that to evaluate the model we perform recursive filtering on the test set to produce states, then use these states to make predictions about observations one time step in the future.

## 6.2 RESULTS

### 6.2.1 ORTHOGONAL RF FOR 2SR

In our first experiment we examine the effectiveness of Orthogonal RF with respect to learning a good PSRNN via 2SR. In figure 3 we compare the MSE for a PSRNN learned via Orthogonal RF with that of one learned using Standard RF for varying numbers of random features. Note that these models were initialized using 2SR but were *not* refined using BPTT. We see that in all cases when the ratio of RF to input dimension is small Orthogonal RF significantly outperforms Standard RF. This difference decreases as the number of RF increases, with both approaches resulting in similar MSE for large RF to input ratios.

### 6.2.2 ORTHOGONAL RF FOR BPTT

In our second experiment we examine the effectiveness of Orthogonal RF with respect to learning a good PSRNN via 2SR initialization *combined with refinement via BPTT*. In figure 4 we compare the MSE for a PSRNN learned via Orthogonal RF with that of one learned using Standard RF over a number of epochs of BPTT. We see that on all datasets, for both Orthogonal RF and Standard RF, MSE decreases as the number of epochs increases. However it is interesting to note that in all datasets Orthogonal RF converges to a better MSE than Standard RF.

---

[2]`https://gym.openai.com/`

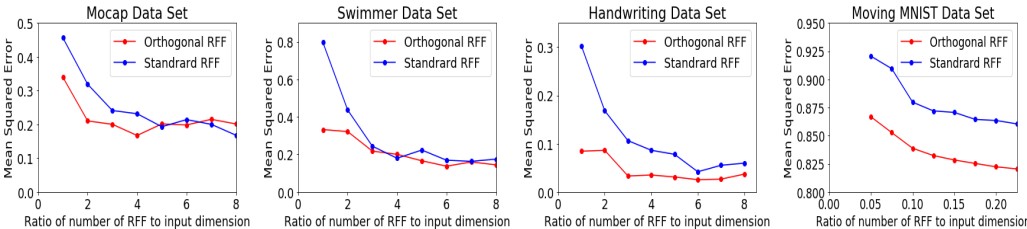

Figure 3: MSE for Orthogonal RF vs Standard RF after two stage regression

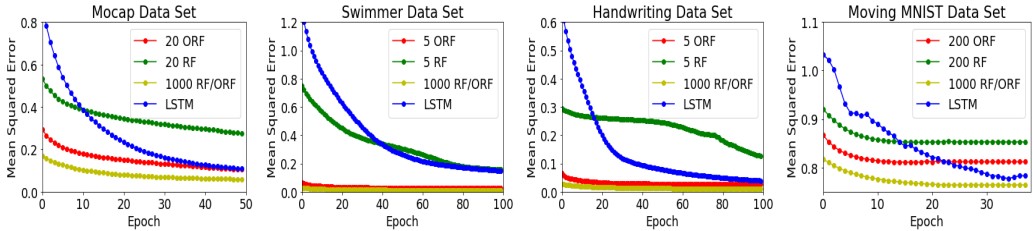

Figure 4: MSE for Orthogonal RF vs Standard RF after two stage regression and BPTT

## 6.3 DISCUSSION

These results demonstrate the effectiveness of Orthogonal RF as a technique for improving the performance of downstream applications. First we have shown that Orthogonal RF can offer significant performance improvements for kernel ridge regression, specifically in the context of the 2SR algorithm for PSRNNs. Furthermore we have shown that not only does the resulting model have lower error, it is also a better initialization for the BPTT gradient descent procedure. In other words, using a model initialization based on orthogonal RF results in BPTT converging to a superior final model.

While the focus of these experiments was to compare the performance of PSRNNs and OPSRNNs, for the sake of completeness we also include error plots for LSTMs. We see that OPSRNNs significantly outperform LSTMs on all data sets.

## 7 CONCLUSIONS

We showed how structured orthogonal constructions can be effectively integrated with recurrent neural network based architectures to provide models that consistently achieve performance superior to the baselines. They also provide significant compression, achieving similar accuracy as PSRNNs with an order of magnitude smaller number of features needed. Furthermore, we gave the first theoretical guarantees showing that orthogonal random features lead to exponentially small bounds on the failure probability regarding empirical risk of the kernel ridge regression model. The latter one is an important component of the RNN based architectures for state prediction that we consider in this paper. Finally, we proved that these bounds are strictly better than for the standard non-orthogonal random feature map mechanism. Exhaustive experiments conducted on several robotics task confirm our theoretical findings.

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

APPENDIX: INITIALIZATION MATTERS: ORTHOGONAL PREDICTIVE STATE RECURRENT NEURAL NETWORKS

We will use the notation from the main body of the paper.

## 8    PROOF OF THEOREM 2

We will assume that a dataset $\mathcal{X} = \{\mathbf{x}_1, ..., \mathbf{x}_N\}$ under consideration is taken from a ball $\mathcal{B}$ of a fixed radius $r$ (that does not depend on data dimensionality $n$ and dataset size $N$) and center $\mathbf{x}_0$. We begin with the following lemma:

**Lemma 1.** *Fix an RBF kernel $K : \mathbb{R}^n \times \mathbb{R}^n$. Consider a randomized kernel estimator $\widehat{K}$ with a corresponding random feature map: $\Phi_{m,n} : \mathbb{R}^n \to \mathbb{R}^{2m}$ and assume that for any fixed $i, j \in \{1, ..., N\}$ the followig holds for any $c > 0$: $\mathbb{P}[|\Phi_{m,n}(\mathbf{x}_i)^\top \Phi_{m,n}(\mathbf{x}_j) - K(\mathbf{x}_i, \mathbf{x}_j)| > c] \leq g(c)$ for some fixed function $g : \mathbb{R} \to \mathbb{R}$. Then with probability at least $1 - N^2 g(c)$, matrix $\widehat{K} + \lambda \mathbf{I}_N$ is a $\Delta$-spectral approximation of matrix $\mathbf{K} + \lambda \mathbf{I}_N$ for $\Delta = \frac{Nc}{\sigma_{\min}}$, where $\sigma_{\min}$ stands for the minimal singular value of $\mathbf{K} + \lambda \mathbf{I}_N$.*

*Proof.* Denote $\mathbf{K} + \lambda N \mathbf{I}_N = \mathbf{V}^\top \boldsymbol{\Sigma}^2 \mathbf{V}$, where an orthonormal matrix $\mathbf{V} \in \mathbf{R}^{N \times N}$ and a diagonal matrix $\boldsymbol{\Sigma} \in \mathbf{R}^{N \times N}$ define the eigendecomposition of $\mathbf{K} + \lambda N \mathbf{I}_N$. Following Avron et al. (2017), we notice that in order to prove that $\widehat{\mathbf{K}} + \lambda \mathbf{I}_N$ is a $\Delta$-spectral approximation of $\mathbf{K} + \lambda \mathbf{I}_N$, it suffices to show that:

$$\|\boldsymbol{\Sigma}^{-1} \mathbf{V} \widehat{\mathbf{K}} \mathbf{V}^\top \boldsymbol{\Sigma}^{-1} - \boldsymbol{\Sigma}^{-1} \mathbf{V} \mathbf{K} \mathbf{V}^\top \boldsymbol{\Sigma}^{-1}\|_2 \leq \Delta. \tag{9}$$

From basic properties of the spectral norm $\|\|_2$ and the Frobenius norm $\|\|_F$ we have:

$$\mathbb{P}[\|\boldsymbol{\Sigma}^{-1} \mathbf{V} \widehat{\mathbf{K}} \mathbf{V}^\top \boldsymbol{\Sigma}^{-1} - \boldsymbol{\Sigma}^{-1} \mathbf{V} \mathbf{K} \mathbf{V}^\top \boldsymbol{\Sigma}^{-1}\|_2 > \Delta] \leq \mathbb{P}[\|\boldsymbol{\Sigma}^{-1} \mathbf{V}\|_2 \|\widehat{\mathbf{K}} - \mathbf{K}\|_F \|\mathbf{V}^\top \boldsymbol{\Sigma}^{-1}\|_2 > \Delta] \tag{10}$$

The latter probability is equal to $p = \mathbb{P}[\|\widehat{\mathbf{K}} - \mathbf{K}\|_F^2 > \frac{\Delta^2}{\|\boldsymbol{\Sigma}^{-1} \mathbf{V}\|_2^2 \cdot \|\mathbf{V}^\top \boldsymbol{\Sigma}^{-1}\|_2^2}]$.

Furthermore, since $\mathbf{V}$ is an isometry matrix, we have: $\|\boldsymbol{\Sigma}^{-1} \mathbf{V}\|_2^2 \leq \frac{1}{\sigma_{\min}}$ and $\|\mathbf{V}^\top \boldsymbol{\Sigma}^{-1}\|_2^2 \leq \frac{1}{\sigma_{\min}}$.

Thus we have:

$$p \leq \mathbb{P}[\|\widehat{\mathbf{K}} - \mathbf{K}\|_F^2 > \Delta^2 \sigma_{\min}^2]. \tag{11}$$

Now notice that from the union bound we get:

$$p \leq N^2 \mathbb{P}[|\widehat{\mathbf{K}}(i,j) - \mathbf{K}(i,j)| > \frac{\Delta \sigma_{\min}}{N}] = N^2 \mathbb{P}[|\Phi_{m,n}(\mathbf{x}_i)^\top \Phi_{m,n}(\mathbf{x}_j) - \mathbf{K}(i,j)| > \frac{\Delta \sigma_{\min}}{N}]. \tag{12}$$

Therefore the probability that $\widehat{\mathbf{K}} + \lambda \mathbf{I}_N$ is a $\Delta$-spectral approximation of $\mathbf{K} + \lambda \mathbf{I}_N$ is at least $1 - N^2 g(c)$ for $c = \frac{\Delta \sigma_{\min}}{N}$ and that completes the proof.

$\square$

Our goal right now is to compute function $g$ from Lemma 1 for random feature maps constructed according to two procedures: the standard one based on independent sampling and the orthogonal one, where marginal distributions corresponding to the joint distribution $(\mathbf{w}_1, ..., \mathbf{w}_m)$ are the same, but vectors $\mathbf{w}_i$ are conditioned to be orthogonal.

We start with a standard random feature map mechanism. Note first that from basic properties of the trigonometric functions we conclude that for any two vectors $\mathbf{x}, \mathbf{y} \in \mathbb{R}^n$, the random feature map approximation of the RBF kernel $K(\mathbf{x}, \mathbf{y})$ which is of the form $\widehat{K}(\mathbf{x}, \mathbf{y}) = |\Phi_{m,n}(\mathbf{x})^\top \Phi_{m,n}(\mathbf{y})$ can be equivalently rewritten as: $\widehat{\mathbf{K}}(\mathbf{x}, \mathbf{y}) = \frac{1}{m} \sum_{i=1}^m \cos(\mathbf{w}_i^\top \mathbf{z})$ for $\mathbf{z} = \mathbf{x} - \mathbf{y}$. This is true for any joint distribution $(\mathbf{w}_1, ..., \mathbf{w}_m)$.

**Lemma 2.** *If mapping $\Phi_{m,n}$ is based on the standard mechanism of independent sampling then one can take as function $g$ from Lemma 1 a function given by the following formula: $g(x) = e^{-Cmx^2}$ for some universal constant $C > 0$.*

*Proof.* Notice first that by the remark above, we get: $\mathbb{P}[|\Phi_{m,n}(\mathbf{x}_i)^\top \Phi_{m,n}(\mathbf{x}_j) - K(\mathbf{x}_i, \mathbf{x}_j)| > x] = \mathbb{P}[\sum_{i=1}^m Z_i > x]$, where $Z_i = \frac{1}{m}\cos(\mathbf{w}_i^\top \mathbf{z}) - \frac{1}{m}\phi(z)$, $z = \|\mathbf{x} - \mathbf{y}\|_2$ and $\phi$ is a positive definite function associated with an RBF kernel $K$. From the unbiasedness of the estimator we see that $\mathbb{E}[Z_i] = 0$. Also, notice that: $|Z_i| \leq \frac{2}{m}$ and different $Z_i$s are independent. Thus, from the standard application of Chernoff inequality we get: $\mathbb{P}[\sum_{i=1}^n Z_i > x] \leq e^{-Cmx^2}$ for some universal constant $C > 0$ (that does not depend on $m$). That completes the proof. $\qquad\square$

By combining Lemma 1 with the formula on $g$ for the standard unstructured case from Lemma 2, we already obtain the formula for $p_{N,m}^{\text{iid}}$ from the statement of the theorem. It remains to show that: $p_{N,m}^{\text{ort}} < p_{N,m}^{\text{iid}}$.

Note that in the previous proof the upper bound on $g$ is derived as a monotonic function of $\mathbb{E}[e^{t\sum_{i=1}^m Z_i}]$ for a parameter $t > 0$ (that is then being optimized),as it is the case for the standard Chernoff's argument. Now, since variables $Z_i$ are independent, we obtained: $\mathbb{E}[e^{t\sum_{i=1}^m Z_i}] = \prod_{i=1}^m \mathbb{E}[e^{tZ_i}]$. Thus, if we can prove that for the continuous-orthogonal distribution we have: $\mathbb{E}[e^{t\sum_{i=1}^m Z_i}] < \prod_{i=1}^m \mathbb{E}[e^{tZ_i}]$, then we complete the proof of Theorem 2 (note that the marginal distributions of $\mathbf{Z}_i$ are the same for both: standard mechanism based on unstructured random features and the one based on continuous-orthogonal sampling of the $m$-tuple of $n$-dimensional vectors).

This is what we prove below.

**Lemma 3.** *Fix some $\mathbf{z} \in \mathbb{R}^n$ and $t > 0$. For a sample $(\mathbf{w}_1^{\text{ort}}, ..., \mathbf{w}_m^{\text{ort}})$ from the continuous-orthogonal distribution the following holds for $n$ large enough:*

$$\mathbb{E}[e^{\frac{t}{n}\sum_{i=1}^m \cos((\mathbf{w}_i^{\text{ort}})^\top \mathbf{z})}] < \prod_{i=1}^m \mathbb{E}[e^{\frac{t}{n}\cos((\mathbf{w}_i^{\text{ort}})^\top \mathbf{z})}]. \tag{13}$$

*Proof.* Since different blocks of vectors $\mathbf{w}_i$ used to construct the orthogonal feature map are independent (the number of blocks is greater than one if $m > n$), it suffices to prove the inequality just for one block. Thus from now on we will focus just on one block and thus without loss of generality we will assume that $m \leq n$.

Note first that

$$\prod_{i=1}^m \mathbb{E}[e^{\frac{t}{n}\cos((\mathbf{w}_i^{\text{ort}})^\top \mathbf{z})}] = \mathbb{E}[e^{\frac{t}{n}\sum_{i=1}^m \cos((\mathbf{w}_i^{\text{iid}})^\top \mathbf{z})}], \tag{14}$$

where $(\mathbf{w}_1^{\text{iid}}, ..., \mathbf{w}_m^{\text{iid}})$ stands for the $m$-tuple sample constructed by the standard unstructured mechanism.

Thus we need to prove that

$$\mathbb{E}[e^{\frac{t}{n}\sum_{i=1}^m \cos((\mathbf{w}_i^{\text{ort}})^\top \mathbf{z})}] < \mathbb{E}[e^{\frac{t}{n}\sum_{i=1}^m \cos((\mathbf{w}_i^{\text{iid}})^\top \mathbf{z})}] \tag{15}$$

$\qquad\square$

Using Taylor expansion for $e^x$, we conclude that it suffices to prove that:

$$\sum_{j_1, j_2, ..., j_m} (\frac{t}{n})^{j_1+...+j_m} \mathbb{E}\Big[\frac{\cos((\mathbf{w}_1^{\text{ort}})^\top \mathbf{z})^{j_1} \cdot ... \cdot \cos((\mathbf{w}_m^{\text{ort}})^\top \mathbf{z})^{j_m}}{j_1! \cdot ... \cdot j_k!}\Big]$$
$$< \sum_{j_1, j_2, ..., j_m} (\frac{t}{n})^{j_1+...+j_m} \mathbb{E}\Big[\frac{\cos((\mathbf{w}_1^{\text{iid}})^\top \mathbf{z})^{j_1} \cdot ... \cdot \cos((\mathbf{w}_m^{\text{iid}})^\top \mathbf{z})^{j_m}}{j_1! \cdot ... \cdot j_k!}\Big], \tag{16}$$

i.e. that:

$$\sum_{j_1, j_2, ..., j_m} (\frac{t}{n})^{j_1+...+j_m} \frac{1}{j_1! \cdot ... \cdot j_m!} \Lambda(j_1, ..., j_m) > 0, \tag{17}$$

where:

$$\Lambda(j_1, ..., j_k) = \mathbb{E}[\cos((\mathbf{w}_1^{\text{iid}})^\top \mathbf{z})^{j_1} \cdot ... \cdot \cos((\mathbf{w}_m^{\text{iid}})^\top \mathbf{z})^{j_m} - \cos((\mathbf{w}_1^{\text{ort}})^\top \mathbf{z})^{j_1} \cdot ... \cdot \cos((\mathbf{w}_m^{\text{ort}})^\top \mathbf{z})^{j_m}] \tag{18}$$

By applying the trigonometric formula:

$$\cos(\alpha)\cos(\beta) = \frac{1}{2}(\cos(\alpha + \beta) + \cos(\alpha - \beta)), \tag{19}$$

we get:

$$\begin{aligned}
\Lambda(j_1, ..., j_k) = \frac{1}{2^{j_1+...+j_m}} \sum_{s_1,...,s_{j_1+...+j_m} \in \{-1,+1\}} \mathbb{E}[ \\
\cos\left(((\mathbf{w}_1^{\text{iid}} \otimes_{s_1} \mathbf{w}_1^{\text{iid}} \otimes_{s_2} ... \otimes_{s_{j_1-1}} \mathbf{w}_1^{\text{iid}}) \otimes_{s_{j_1}} ...)^\top \mathbf{z}\right) - \\
\cos\left(((\mathbf{w}_1^{\text{ort}} \otimes_{s_1} \mathbf{w}_1^{\text{ort}} \otimes_{s_2} ... \otimes_{s_{j_1-1}} \mathbf{w}_1^{\text{ort}}) \otimes_{s_{j_1}} ...)^\top \mathbf{z}\right)],
\end{aligned} \tag{20}$$

where $\otimes_1$ stands for vector-addition operator and $\otimes_{-1}$ stands for vector-subtraction operator.

Note that without loss of generality we can assume that $s_{j_1} = s_{j_1+j_2} = ... = +1$, since for other configurations we obtain a random variable of the same distribution. Consider a fixed configuration $(s_1, s_2, ..., s_{j_1+...+j_m})$ and the corresponding term of the sum above that can be rewritten in the compressed way as:

$$F = \mathbb{E}[\cos(n_1\mathbf{w}_1^{\text{iid}} + n_2\mathbf{w}_2^{\text{iid}} + ... + n_m\mathbf{w}_m^{\text{iid}})^\top \mathbf{z}] - \mathbb{E}[\cos(n_1\mathbf{w}_1^{\text{ort}} + n_2\mathbf{w}_2^{\text{ort}} + ... + n_m\mathbf{w}_m^{\text{ort}})^\top \mathbf{z}], \tag{21}$$

for some $n_1, ..., n_m \in \mathbb{Z}$. Without loss of generality, we can assume that $n_1, ..., n_m \in \mathbb{N}$, since the distribution of the random variables under consideration does not change if $n_i$ is replaced with $-n_i$. Without loss of generality we can also assume that there exists $i \in \{1, ..., m\}$ such that $n_i > 0$, since otherwise the corresponding term $F$ is equal to 0.

Denote by $R_1, ..., R_m$ the set of independent random variables, where each is characterized by the distribution which is the distribution of vectors $\mathbf{w}_i^{\text{iid}}$ (and thus also of vectors $\mathbf{w}_i^{\text{ort}}$). Denote: $R = \sqrt{n_1^2 R_1^2 + ... + n_m^2 R_m^2}$. Note that $n_1\mathbf{w}_1^{\text{ort}} + n_2\mathbf{w}_2^{\text{ort}} + ... + n_m\mathbf{w}_m^{\text{ort}} \sim R\mathbf{v}$, where $\mathbf{v}$ is a unit $L_2$-norm vector taken uniformly at random from the sphere of radius 1 and furthermore: $R$ and $\mathbf{v}$ are chosen independently. That is implied by the isotropicity of vectors $\mathbf{w}_i^{\text{ort}}$. Similarly, denote $\widehat{R} = \sqrt{R^2 + \sum_{i,j \in \{1,...,m\}} n_i n_j R_i R_j \mathbf{v}_i^\top \mathbf{v}_j}$, where $\mathbf{v}_1, ..., \mathbf{v}_m$ stand for the independent copies of $\mathbf{v}$. Note that, by the similar analysis as before, we conclude that $n_1\mathbf{w}_1^{\text{iid}} + n_2\mathbf{w}_2^{\text{iid}} + ... + n_m\mathbf{w}_m^{\text{iid}} \sim \widehat{R}\mathbf{v}$ and furthermore, $\widehat{R}$ and $\mathbf{v}$ are chosen independently.

Therefore, by expanding $\cos(x)$ using Taylor expansion, we get:

$$F = \sum_{k=0}^{\infty} \frac{\|\mathbf{z}\|^{2k}(-1)^k \mathbb{E}[(\mathbf{v}^\top \widehat{\mathbf{z}})^{2k}]}{(2k)!} \mathbb{E}[\widehat{R}^{2k}] - \sum_{k=0}^{\infty} \frac{\|\mathbf{z}\|^{2k}(-1)^k \mathbb{E}[(\mathbf{v}^\top \widehat{\mathbf{z}})^{2k}]}{(2k)!} \mathbb{E}[R^{2k}], \tag{22}$$

where: $\widehat{\mathbf{z}} = \frac{\mathbf{z}}{\|\mathbf{z}\|}$. Denote: $A(k,n) = \mathbb{E}[(\mathbf{v}^\top \widehat{\mathbf{z}})^k]$ (note that $\mathbf{v}, \widehat{\mathbf{z}} \in \mathbb{R}^n$). It is easy to see that for odd $k$ we have: $A(n,k) = 0$. We obtain:

$$F = \sum_{k=0}^{\infty} \frac{\|\mathbf{z}\|^{2k}(-1)^k A(2k,n)}{(2k)!}(\mathbb{E}[\widehat{R}^{2k}] - \mathbb{E}[R^{2k}]). \tag{23}$$

The following technical fact will be useful:

**Lemma 4.** *Expression $A(k,n)$ is given by the following formula:*

$$A(k,n) = \frac{1}{\int_0^\pi \sin^{n-2}(\theta)d\theta} \int_0^\pi \cos^k(\theta)\sin^{n-2}(\theta)d\theta, \tag{24}$$

*which can be equivalently rewritten (by computing the integrals explicitly) as:*

$$A(2k,n) = \frac{(n-2)(n-4)\cdot...\cdot\delta(n=2)}{(n-3)(n-5)\cdot...\cdot\gamma(n=2)}\cdot\frac{(2k-1)!!}{(n-1)(n+1)...(n+2k-3)}\cdot$$
$$\frac{(n+2k-3)(n+2k-5)...\cdot\gamma(n=2)}{(n+2k-2)(n+2k-4)...\cdot\delta(n=2)},\tag{25}$$

*where:* $\delta(n=2)=2$ *if* $n=2$ *and* $\delta(n=2)=1$ *otherwise and:* $\gamma(n=2)=1$ *if* $n=2$ *and* $\gamma(n=2)=2$ *otherwise.*

In particular, the following is true:

$$|A(2k,n)| \le \frac{(2k-1)!!}{(n-1)(n+1)\cdot...\cdot(n+2k-3)}.\tag{26}$$

We will use that later. Note that $\mathbf{v}_i^\top\mathbf{v}_j \sim \mathbf{v}^\top\widehat{\mathbf{z}}$. Therefore we obtain:

$$F = \sum_{k=0}^{\infty}\frac{\|\mathbf{z}\|^{2k}(-1)^k A(2k,n)}{(2k)!}\alpha_k,\tag{27}$$

where

$$\alpha_k = \sum_{i=1}^{k}\binom{k}{i}\mathbb{E}[(R^2)^{k-i}\lambda^i],\tag{28}$$

and $\lambda = \sum_{i,j\in\{1,...,m\}} n_i n_j R_i R_j \mathbf{v}_i^\top\mathbf{v}_j$.

Note that $\mathbb{E}[(R^2)^{k-1}\lambda] = 0$ since $\mathbb{E}[(\mathbf{v}_i^\top\mathbf{v}_j)] = 0$ and furthermore, directions $\mathbf{v}_1,...,\mathbf{v}_m$ are chosen independently from lengths $R_1,...,R_m$. Therefore we have:

$$\alpha_k = \binom{k}{2}\mathbb{E}[(R^2)^{k-2}\lambda^2] + \beta_k,\tag{29}$$

where:

$$\beta_k = \sum_{i=3}^{k}\binom{k}{i}\mathbb{E}[(R^2)^{k-i}\lambda^i].\tag{30}$$

Now let us focus on a single term $\rho = \mathbb{E}[(R^2)^{k-l}\lambda^l]$ for some fixed $l \ge 3$.

Note that the following is true:

$$\rho \le \mathbb{E}[(R^2)^{k-l}(\sum_{i,j\in\{1,...,m\}} n_i n_j R_i R_j)^l] \cdot \max_{i_1,j_1,...,i_l,j_l}\mathbb{E}[|\mathbf{v}_{i_1}^\top\mathbf{v}_{j_1}| \cdot ... \cdot |\mathbf{v}_{i_l}^\top\mathbf{v}_{j_l}|],\tag{31}$$

where the maximum is taken over $i_1,j_1,...,i_l,j_l \in \{1,...,m\}$ such that $i_s \ne j_s$ for $s = 1,...,l$.

Note first that:

$$\mathbb{E}[(R^2)^{k-l}(\sum_{i,j\in\{1,...,m\}} n_i n_j R_i R_j)^l] \le$$

$$\mathbb{E}[(R^2)^{k-l}(\sum_{i,j\in\{1,...,m\}}\frac{(n_i R_i)^2 + (n_j R_j)^2}{2})^l] \le \mathbb{E}[(R^2)^{k-l}(m-1)^l(R^2)^l] \le (m-1)^l\mathbb{E}[R^{2k}].$$

$$\tag{32}$$

Let us focus now on the expression $\max_{i_1,j_1,...,i_l,j_l}\mathbb{E}[|\mathbf{v}_{i_1}^\top\mathbf{v}_{j_1}| \cdot ... \cdot |\mathbf{v}_{i_l}^\top\mathbf{v}_{j_l}|]$.

We will prove the following upper bound on $\max_{i_1,j_1,...,i_l,j_l}\mathbb{E}[|\mathbf{v}_{i_1}^\top\mathbf{v}_{j_1}| \cdot ... \cdot |\mathbf{v}_{i_l}^\top\mathbf{v}_{j_l}|]$.

**Lemma 5.** *The following is true:*

$$\mathbb{E}[|\mathbf{v}_{i_1}^\top \mathbf{v}_{j_1}| \cdot \ldots \cdot |\mathbf{v}_{i_l}^\top \mathbf{v}_{j_l}|] \leq (\frac{\log(n)}{\sqrt{n - \sqrt{n}\log(n)}})^l + l(e^{-\frac{\log^2(n)}{4}} + e^{-\frac{\log^2(n)}{2}}). \tag{33}$$

*Proof.* Note that from the isotropicity of Gaussian vectors we can conclude that each single $|\mathbf{v}_{i_s}^\top \mathbf{v}_{j_s}|$ is distributed as: $\frac{g_1}{\sqrt{g_1^2 + \ldots + g_n^2}}$, where $g_1, \ldots, g_n$ stand for $n$ independent copies of a random variable taken from $\mathcal{N}(0,1)$. Note that $g_1^2 + \ldots + g_n^2$ is taken from the $\chi_n^2$- distribution. Using the well-known bounds for the tails of $\chi_n^2$- distributions, we get: $\mathbb{P}[g_1^2 + \ldots + g_n^2 - n \leq a] \leq e^{-\frac{a^2}{4n}}$. Note also that $\mathbb{P}[|g_1| > x] \leq 2\frac{e^{-\frac{x^2}{2}}}{x\sqrt{2\pi}}$. Thus, by taking: $a = \sqrt{n}\log(n)$, $x = \log(n)$ and applying union bound, we conclude that with probability at least $1 - e^{-\frac{\log^2(n)}{4}} - e^{-\frac{\log^2(n)}{2}}$ a fixed random variable $|\mathbf{v}_{i_s}^\top \mathbf{v}_{j_s}|$ satisfies: $|\mathbf{v}_{i_s}^\top \mathbf{v}_{j_s}| \leq \frac{\log(n)}{\sqrt{n - \sqrt{n}\log(n)}}$. Thus, by the union bound we conclude that for any fixed $i_1, j_1, \ldots, i_l, j_l$ random variable $|\mathbf{v}_{i_1}^\top \mathbf{v}_{j_1}| \cdot \ldots \cdot |\mathbf{v}_{i_l}^\top \mathbf{v}_{j_l}|$ satisfies: $|\mathbf{v}_{i_1}^\top \mathbf{v}_{j_1}| \cdot \ldots \cdot |\mathbf{v}_{i_l}^\top \mathbf{v}_{j_l}| \leq (\frac{\log(n)}{\sqrt{n - \sqrt{n}\log(n)}})^l$ with probability at least $1 - l(e^{-\frac{\log^2(n)}{4}} + e^{-\frac{\log^2(n)}{2}})$. Since $|\mathbf{v}_{i_1}^\top \mathbf{v}_{j_1}| \cdot \ldots \cdot |\mathbf{v}_{i_l}^\top \mathbf{v}_{j_l}|$ is upper bounded by one, we conclude that:

$$\mathbb{E}[|\mathbf{v}_{i_1}^\top \mathbf{v}_{j_1}| \cdot \ldots \cdot |\mathbf{v}_{i_l}^\top \mathbf{v}_{j_l}|] \leq (\frac{\log(n)}{\sqrt{n - \sqrt{n}\log(n)}})^l + l(e^{-\frac{\log^2(n)}{4}} + e^{-\frac{\log^2(n)}{2}}). \tag{34}$$

$\square$

Using Lemma 5, we can conclude that:

$$\rho \leq (m-1)^l \mathbb{E}[R^{2k}] \cdot \left( (\frac{\log(n)}{\sqrt{n - \sqrt{n}\log(n)}})^l + l(e^{-\frac{\log^2(n)}{4}} + e^{-\frac{\log^2(n)}{2}}) \right) \tag{35}$$

Therefore we have:

$$\beta_k \leq \sum_{i=3}^k \binom{k}{i}(m-1)^i \mathbb{E}[R^{2k}] \cdot \left( (\frac{\log(n)}{\sqrt{n - \sqrt{n}\log(n)}})^i + i(e^{-\frac{\log^2(n)}{4}} + e^{-\frac{\log^2(n)}{2}}) \right) \tag{36}$$

Thus we can conclude that for $n$ large enough:

$$\beta_k \leq k(2m)^k (\frac{2\log^3(n)}{n^{\frac{3}{2}}} + 2ke^{-\frac{\log^2(n)}{4}})\mathbb{E}[R^{2k}]. \tag{37}$$

Thus we get:

$$F = \sum_{k=0}^\infty \frac{\|\mathbf{z}\|^{2k}(-1)^k A(2k,n)}{(2k)!}\binom{k}{2}\mathbb{E}[(R^2)^{k-2}\lambda^2] + \Gamma, \tag{38}$$

where:

$$|\Gamma| \leq \sum_{k=0}^\infty \frac{\|\mathbf{z}\|^{2k}A(2k,n)}{(2k)!}k(2m)^k(\frac{2\log^3(n)}{n^{\frac{3}{2}}} + 2ke^{-\frac{\log^2(n)}{4}})\mathbb{E}[R^{2k}]$$
$$= \frac{2\log^3(n)}{n^{\frac{3}{2}}}A + 2e^{-\frac{\log^2(n)}{4}}B, \tag{39}$$

$B$ satisfies: $B = Ak$ and $A$ is given as:

$$A = \sum_{k=0}^\infty \frac{\|\mathbf{z}\|^{2k}\mathbb{E}[R^{2k}]A(2k,n)}{(2k)!}k(2m)^k. \tag{40}$$

Now note that since data is taken from the ball of radius $r$, we have: $\|\mathbf{z}\| \leq 2r$. Furthermore, from the smoothness of the considered class of RBF kernels, we obtain:

$$\mathbb{E}[R_{2k}] \leq \max_{i=1,\ldots,m} n_i^{2k} m^k (n-1)(n+1) \cdots (n+2k-3) f^k(k) k!. \tag{41}$$

Denote $h = \operatorname{argmax}_{i=1,\ldots,m} n_i$. Note that $(2k-1)!! = \frac{(2k)!}{2^k k!}$. Thus, by applying the above upper bound on $A(2k, n)$, we obtain:

$$A, B \leq \sum_{k=0}^{\infty} (m^2 (2r)^2 n_h^2 f(k))^k k^2 \leq \sum_{k=0}^{\infty} (4m^2 (2r)^2 n_h^2 f(k))^k. \tag{42}$$

Now, notice that for a given $n_h$, $m$ and $r$, the above upper bound is of the form $\sum_{k=0}^{\infty} q_k^k$, where $q_k \to 0$ as $k \to \infty$ (since $f_k \to 0$ as $k \to \infty$ for smooth RBF kernels). Then we can conclude that the contribution of the terms $\Gamma$ to the expression $\tau = \frac{\Lambda(j_1,\ldots,j_m)}{j_1! \cdots j_m!}$ from the LHS of 17 is of the order $O(\frac{1}{n^{\frac{3}{2}}})$ (notice that every $n_i$ satisfies: $n_i \leq j_i$). Now let us consider $F - \Gamma$. We have:

$$F - \Gamma = \sum_{k=0}^{\infty} \frac{\|\mathbf{z}\|^{2k} (-1)^k A(2k, n)}{(2k)!} \binom{k}{2} \mathbb{E}[(R^2)^{k-2} \lambda^2] \tag{43}$$

By the same analysis as before we conclude that

$$F - \Gamma = \rho + o_n(\frac{1}{n}), \tag{44}$$

where

$$\rho = \sum_{k=0}^{\infty} \frac{\|\mathbf{z}\|^{2k} (-1)^k A(2k, n)}{(2k)!} \binom{k}{2} \mathbb{E}[(\widehat{R}^2)^{k-2} \lambda^2] \tag{45}$$

and futhermore: $\rho = \frac{1}{n} \tilde{\rho} + o_n(\frac{1}{n})$, where

$$\tilde{\rho} = W \sum_{k=0}^{\infty} \frac{\|\mathbf{z}\|^{2k} (-1)^k A(2k, n)}{(2k)!} \binom{k}{2} \mathbb{E}[(\widehat{R}^{2k})] \tag{46}$$

and $W$ is some constant (that depends on $m$). Thus to show Inequality 17 for $n$ large enough, it suffices to show that $\tilde{\rho} > 0$ and that $\rho$ does not depend on $n$ (even though $n$ is explicitly embedded in the formula on $\tilde{\rho}$ via $A(2k, n)$). This follows from the straightforward extension (for different $n_i$s) of the fact that for $n_1 = \ldots = n_m$, $\tilde{\rho}$ can be rewritten in terms of the positive definite function $\phi$ describing an RBF kernel under consideration, namely:

$$\tilde{\rho} = \frac{1}{8n} ((\|n_1 \mathbf{z}\|^2 \frac{d^2(\phi^m(x))}{dx^2})_{|x=\|n_1 \mathbf{z}\|} - (\|n_1 \mathbf{z}\| \frac{d(\phi^m(x))}{dx})_{|x=\|n_1 \mathbf{z}\|}). \tag{47}$$

That the above expression is positive is implied by the fact that every positive definite function $\phi$ (not parametrized by $n$) can be rewritten as $\phi(r) = \sigma(r^2)$, where $\sigma$ is completely monotone on $[0, +\infty]$ and from the basic properties of completely monotone functions (see: Schoenberg (1938)). That completes the proof of the theorem.

## 9 PROOF OF THEOREM 3

The theorem follows straightforwardly from Theorem 1, Theorem 2 and the fact that $\operatorname{rank}(\widehat{\mathbf{K}}) \leq m$ (since $m$-dimensional random feature maps are used for kernel approximation).

