# OpenReview forum: "Initialization matters: Orthogonal Predictive State Recurrent Neural Networks"
_ICLR.cc/2018/Conference — Accept (Poster)_

### Official Review · AnonReviewer2 · 2017-11-26
**The aim of the paper is to improve the performances of  Predictive State Recurrent Neural Networks (PSRNNs) by considering Orthogonal Random Features instead of Random Features as roginatgly done in the seminal work of Downey et al. 2017.**

**Rating:** 4
**Confidence:** 5

**Review:**

I was very confused by some parts of the paper that are simple copy-past from the paper of Downey et al.  which has been accepted for publication in NIPS. In particular, in section 3, several sentences are taken as they are from the Downey et al.’s paper. Some examples :

« provide a compact representation of a dynamical system
by representing state as a set of predictions of features of future observations. »

« a predictive state is defined as… , where…  is a vector of features of future observations and ...  is a vector of
features of historical observations. The features are selected such that ...  determines the distribution
of future observations … Filtering is the process of mapping a predictive state… »
Even the footnote has been copied & pasted: « For convenience we assume that the system is k-observable: that is, the distribution of all future observations
is determined by the distribution of the next k observations. (Note: not by the next k observations
themselves.) At the cost of additional notation, this restriction could easily be lifted. »
«  This approach is fast, statistically consistent, and reduces to simple
linear algebra operations. »

Normally, I should have stopped reviewing, but I decided to continue  since those parts only concerned the preliminaries part.

A key element in PSRNN is to used as an initialization a kernel ridge regression. The main result here, is to show that using orthogonal random features approximates well the original kernel comparing to random fourrier features as considered in PSRNN. This result is formally stated and proved in the paper.

The paper comes with some experiments in order to empirically demonstrate the superiority  orthogonal random features over RFF. Three data sets are considered (Swimmer,  Mocap and  Handwriting).

I found it that the contribution of the paper is very limited. The connexion to PSRNN is very tenuous since the main results are about the regression part. in Theorems 2 and 3 there are no mention to PSRNN.

Also the experiment is not very convincing. The datasets are too small with observations in low dimensions, and I found it not very fair to consider LSTM in such settings.

Some minor remarks:

- p3: We use RFs-> RFFs
- p5: ||X||, you mean |X| the size of the dataset
- p12: Eq (9). You need to add « with probability $1-\rho$ as in Avron’s paper.
- p12: the derivation of Eq (10) from Eq (9) needs to be detailed.


I thank the author for their detailed answers. Some points have been clarified but other still raise issues. In particular, I continue thinking that the contribution is limited. Accordingly, I did not change my scores.

---

### Official Review · AnonReviewer1 · 2017-11-27
**new result on risk of KRR using orthogonal features, applied to fast training of predictive state RNNs**

**Rating:** 8
**Confidence:** 4

**Review:**

The paper tackles the problem of training predictive state recurrent neural networks (PSRNN), which
uses large kernel ridge regression (KRR) problems as a subprimitive, and makes two main contributions:
- the suggestion to use orthogonal random features (ORFs) in lieu of standard random fourier features (RFFs) to reduce the size of the KRR problems
- a novel analysis of the risk of KRR using ORFs which shows that the risk of ORFs is no larger than that of using RFFs

The contribution to the practice of PSRNNs seems significant (to my non-expert eyes): when back-propagation through time is used, using ORFs to do the two-stage KRR training needed visibly outperforms using standard RFMs to do the KRR. I would like the authors to have provided results on more than the current three datasets, as well as an explanation of how meaningful the MSEs are in each dataset (is a MSE of 0.2 meaningful for the Swimmer Dataset, for instance? the reader does not know apriori).

The contribution in terms of the theory of using random features to perform kernel ridge regression is novel, and interesting. Specifically, the author argue that the moment-generating function for the pointwise kernel approximation error of ORF features grows slower than the moment-generating function for the pointwise kernel approximation error of RFM features, which implies that error bounds derived using the MGF of the RFM features will also hold for ORF features. This is a weaker result than their claim that ORFs satisfy better error, but close enough to be of interest and certainly indicates that their method is principled. Unfortunately, the proof of this result is poorly written:
- equation (20) takes a long time to parse --- more effort should be put into making this clear
- give a reference for the expressions given for A(k,n) in 24 and 25
- (27) and (28) should be explained in more detail.
My staying power was exhausted around equation 31. The proof should be broken up into several manageable lemmas instead of its current monolithic and taxing form.

---

### Official Review · AnonReviewer3 · 2017-11-28
**Nice theory but small empirical evaluation**

**Rating:** 7
**Confidence:** 2

**Review:**

This paper investigates the Predictive State Recurrent Neural Networks (PSRNN) model that embed the predictive states in a Reproducible Hilbert Kernel Space and then update the predictive states given new observation in this space.
While PSRNN usually uses random features to project the map the states in a new space where dot product approximates the kernel well, the authors proposes to leverage orthogonal random features.

In particular, authors provide theoretical guarantee and show that the model using orthogonal features has a smaller upper bound on the failure probability regarding the empirical risk than the model using unstructured randomness.

Authors then empirically validate their model on several small-scale datasets where they compare their model with PSRNN and LSTM. They observe that PSRNN with orthogonal random features leads to lower MSE on test set than both PSRNN and LSTM and seem to reach lower value earlier in training.

Question:
-	What is the cost of constructing orthogonal random features compared to RF?
-	What is the definition of H the Hadamard matrix in the discrete orthogonal joint definition?
-	What are the hyperparameters values use for the LSTM
-	Empirical evaluations seem to use relatively small datasets composed by few dozens of temporal trajectories. Did you consider larger dataset for evaluation?
-	How did you select the maximum number of epochs in Figure 5? It seems that the validation error is still decreasing after 25 epochs?

Pros:
-	Provide theoretical guarantee for the use of orthogonal random features in the context of PSRNN
Cons:
-	Empirical evaluation only on small scale datasets.

---

### Decision · Program_Chairs · 2018-01-29
**ICLR 2018 Conference Acceptance Decision**

**Decision:**

Accept (Poster)

**Comment:**

this submission presents the positive impact of using orthogonal random features instead of unstructured random features for predictive state recurrent neural nets. there's been some sentiment by the reviewers that the contribution is rather limited, but after further discussion with another AC and PC's, we have concluded that it may be limited but a solid follow-up on the previous work on predictive state RNN.